# Severe Drought in the Spring of 2020 in Poland—More of the Same?

**Iwona Pińskwar \*** , **Adam Choryński** and **Zbigniew W. Kundzewicz**

Institute for Agricultural and Forest Environment, Polish Academy of Sciences, Bukowska 19, 60-809 Poznań, Poland; adam.chorynski@isrl.poznan.pl (A.C.); kundzewicz@yahoo.com (Z.W.K.)

\* Correspondence: iwona.pinskwar@isrl.poznan.pl

**Abstract:** Two consecutive dry years, 2018 and 2019, a warm winter in 2019/20, and a very dry spring in 2020 led to the development of severe drought in Poland. In this paper, changes in the Standardized Precipitation-Evapotranspiration Index (SPEI) for the interval from 1971 to the end of May 2020 are examined. The values of SPEI (based on 12, 24 and 30 month windows, i.e., SPEI 12, SPEI 24 and SPEI 30) were calculated with the help of the Penman–Monteith equation. Changes in soil moisture contents were also examined from January 2000 to May 2020, based on data from the NASA Goddard Earth Sciences Data and Information Services Center, presenting increasing water shortages in a central belt of Poland. The study showed that the 2020 spring drought was among the most severe events in the analyzed period and presented decreasing trends of SPEI at most stations located in central Poland. This study also determined changes in soil moisture contents from January 2000 to May 2020 that indicate a decreasing tendency. Cumulative water shortages from year to year led to the development of severe drought in the spring of 2020, as reflected in very low SPEI values and low soil moisture.

**Keywords:** standardized precipitation-evapotranspiration index; soil moisture; Poland

## 1. Introduction

Droughts, in addition to floods, are extreme weather events of utmost importance in the world. Scarcity of precipitation, exacerbated by high air temperature, may lead to the development of different stages of droughts: from meteorological, through soil to hydrological droughts [1]. As droughts develop, their effects on agriculture, water supply, energy and the environment grow. Since the year 2000, Europe has faced multiple severe droughts, accompanied by heat waves, such as in 2003, 2006, 2010, 2015, 2018, 2019 and 2020 [2,3], as well as extensive forest fires [4].

Droughts heavily impact many sectors of the national economy and agriculture in particular [5]. Analyses show that yield losses in Poland may even reach 50% on light soils [6]. A drought that took place in 2015 in Poland caused 12% yield loss of cereals, 20% of vegetables, 25% of potatoes and 50–70% of silage maize [5]. However, in other countries, the agricultural sector has suffered even more than in Poland. For example, crop damage extent in two dry periods (2004–2005 and 2011–2012) reached 70–95% in southwestern Spain [7]. In the summer of 2018, an extreme drought occurred in central and northern Europe [8]. Economic costs of agricultural crop losses caused by this event, which can be estimated based on values of compensations to farmers, reached €340 million in Germany and €116 million in Sweden [9].

Looking at the impacts of droughts, researchers also investigate the more general problem of their influence on the economy. Analyses combining the future impact of droughts with projected temperature increase assess that in over 75 countries, the gross domestic product will be adversely affected by a 1.5 °C temperature rise scenario [10]. The same authors claim that in this scenario,

historical 50 year drought frequency will double across 58% of global land areas. Moreover, at a 2 °C temperature rise scenario, 92 countries will face dramatic droughts, significantly impacting their economy, and 67% of land areas may be impacted by catastrophic droughts. One should bear in mind that droughts influence countries with a lower economic resilience more, and countries' resilience is correlated with their wealth. Therefore, poorer countries and their populations are more vulnerable to droughts. It is also important to note that drought events and their consequences are more probable to affect populations in the tropical and mid-latitude regions [10].

The unusual 2003 drought, which had severe impacts in much of Europe, has been seen by researchers as the "shape of things to come", because such extreme temperatures as observed during that summer are projected to occur much more frequently at the end of the 21st century [11].

The drought and heat waves in 2010 were exceptionally severe in eastern Europe and large parts of Russia. According to Barriopedro et al. [12], the amplitude and spatial extent of this anomalous hot spell exceeded the amplitude and spatial extent of the previous hottest summer of 2003. These two "mega-heatwaves" in 2003 and 2010 probably broke the 500 year temperature records over approximately 50% of Europe [12].

The next combination of drought and heat waves recorded in Europe, particularly in the central and eastern part of the continent, in 2015, was the third (after 2003 and 2010) warmest summer in Europe since 1910 [2]. During this drought, extreme low flows were recorded in several rivers, especially in the central and eastern parts of Europe [13]. In Poland, too, the extremely hot and dry summer of 2015 led to the lowest values of the stages (and discharges) on record, for example in the Vistula River where the stage was the lowest since the 18th century, when records began [14].

The 2015 drought was extreme, but not singular. In the spring of 2018, a longer drought commenced, covering two years, 2018 and 2019. Hari et al. [3] showed that the occurrence of this drought was unprecedented in the last 250 years, and its impact on vegetation was more severe than that of the 2003 European drought.

The water resources in Poland should recover during winter, when evapotranspiration is smaller and, typically, snow cover exists. However, over several years now, winters in Poland have been frequently warmer than the long-term average [15]. Additionally, in recent years snow cover has been less abundant to non-existent in much of Poland. According to Copernicus Climate Change Service/ECMWF (European Centre for Medium-Range Weather Forecasts), the winter of 2019–2020 was the warmest winter season ever recorded in Europe [16].

Out of the last five years, three have been dry (2015, 2018, 2019), with annual precipitation totals in Poznań being, respectively: 437.7 mm, 372.5 mm and 392.8 mm. These values are well below the long-term average for the period 1951–2019, which is 521.6 mm. The cumulative two-year rainfall deficit (the sum of the annual shortages in relation to the average) in 2018 and 2019 was equal to 277.9 mm. However, the low rainfall in 2018–2019 was not something exceptional, because in 1982 only 275 mm were recorded in Poznań, and in the next, only slightly less dry year 1983, the rainfall was 333.5 mm. Thus, the two-year rainfall deficit reached a record value of 412.7 mm in 1982–1983 (data from IMGW-PIB).

In Poland, after the severe 2018–2019 drought, the precipitation total during the autumn of 2019 and the winter of 2019–2020 was also near normal of the long-term mean (e.g., for the period 1981–2010). This did not improve the hydrological situation [15]. By contrast, precipitation in February 2020 was extremely high in Poland. For many stations, monthly precipitation anomalies, related to the mean for 1981–2010, were above 200%; and nearly 280% (76.3 mm—an all-time record for February) in Poznań. However, it was rainfall rather than snowfall. Much of this rain was intense, so that surface runoff was high. Additionally, February 2020 was extremely warm [17] and evapotranspiration was high as well. In turn, spring 2020 was very dry. In consequence, after the very dry years of 2018–2019, rather dry cold months (except for February 2020) and the very dry spring of 2020 (especially April), extreme drought developed. Figure 1 shows water scarcity on May 21, 2020, with a depth from 7 to 28 cm.

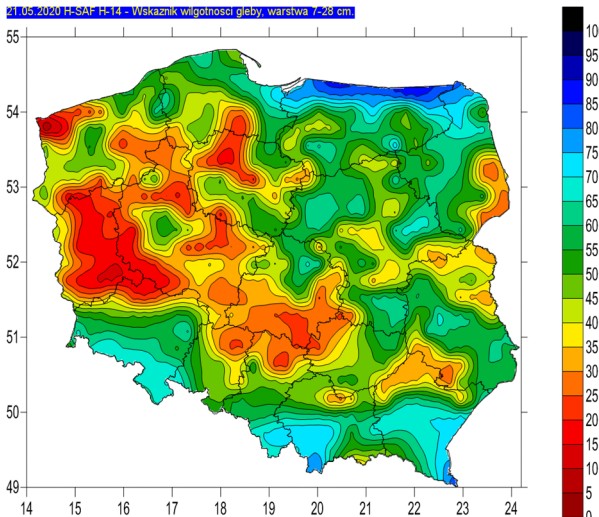

**Figure 1.** Spatial distribution of soil moisture (expressed as a percentage) in Poland, in the subsurface (depth from 7 to 28 cm) on 21 May 2020, created by EUMETSAT H-SAF, based on satellite data (ASCAT sensor, Metop satellites), using the ECMWF H-TESSEL model, with a spatial resolution of 25 km. Source: http://stopsuszy.imgw.pl/wilgotnosc/.

In the spring of 2020, in consequence of the severe and long-lasting drought, a large (5280 hectares) wildfire took place in the Biebrza National Park. Nearly 10% of the whole area of the park burned [18]. Apart from large losses in plants and animals, the firefighting operation incurred high costs. They were estimated by the Ministry of Environment at the level of 8 million PLN (almost €2 million) [19]. About 1.5 thousand firefighters were engaged, as well as more than 300 other rescuers and soldiers. Moreover, six airplanes and two helicopters supported the action.

Regional multi-model experiments showed that the frequency of occurrence of mega-heatwaves during summers may increase by a factor of 5 to 10 within the next 40 years [12]. The higher the atmospheric concentration of greenhouse gases, the greater will the role of anthropogenic warming in exacerbating the future risk of multi-year drought events be. Climate model simulations under the highest Representative Concentration Pathway (RCP8.5) predict a seven-fold increase in the occurrence of the consecutive droughts during the second half of the 21st century [3]. If the rate of anthropogenic warming slows down, the risk of future droughts will be reduced [3]. In addition, Samaniego et al. [20] pointed out that a global warming of 3 K will lead to increase in droughts: an event like the 2003 drought is projected to become twice as frequent in Europe.

In Poland, where rain-fed agriculture has been prevailing, soil moisture deficits are the main factor adversely affecting the agricultural productivity. Droughts as severe as those that occurred in Poland in 2015, 2018 and 2019 are projected to be more frequent in the future and to affect larger areas [21].

Due to low mean annual precipitation (a national average of 624 mm) [22] and low water resources (a mean of about 60 billion $m^3$, and in dry years much less: below 40 billion $m^3$), the per capita water availability in Poland is far below the European average. It is only possible to store approximately 6.5% of annual river flow in reservoirs in Poland [23], hence there would be no water for massive agricultural irrigation in the country, which may be needed in the warming climate. Water stress is on the rise in Poland; hence drought analysis is of much relevance and interest in the nation.

Problems with water scarcity in Poland (resulting from low precipitation and availability of renewable water resources, in particular per capita) are serious in the whole country, but they are most severe for lowland areas in central Poland. The long-term mean annual precipitation totals are small in that region, between 500 and 600 mm with deficit from the norm (e.g., the annual precipitation observed in 1982 in Poznan was 275 mm only; it was even lower in Kalisz in 2015: 259 mm) [24], causing water problems; hence this paper focuses on this area. In this study, the

Standardized Precipitation-Evapotranspiration Index (SPEI) was used for station data for the period from January 1971 to May 2020.

The SPEI is based on the difference between precipitation and potential evapotranspiration. These anomalies in climatic water balance are normalized, which allows to determine the onset, duration and magnitude of drought conditions [25]. The potential evapotranspiration may be calculated using different methods, such as the Thornthwaite equation [26], which is used to estimate the SPEI in the global drought monitor [27], and the Penman–Monteith method, which is recommended by the FAO (Food and Agriculture Organization of the United Nations). The new SPEI base, which uses the FAO-56 Penman–Monteith estimation, is based on monthly precipitation and potential evapotranspiration from the Climatic Research Unit of the University of East Anglia and is updated as soon as new data are available, currently up to December 2018.

In Poland, different studies analyzed the SPEI based on the Thornthwaite equation. It was investigated based on station data by Wibig [28] and based on gridded data by Somorowska [14]. Wibig [28] analyzed drought events during the period of 1951–2006 for 18 stations on five time scales: lasting from 1 to 24 months. She found that all trends were decreasing, but statistically insignificant. By contrast, Somorowska [14] analyzed data from 1956 to the end of 2015, showing a statistically significant decreasing trend in the SPEI for a relatively large area of Poland from the southwest towards the central part of the country.

The novelty of the current research is that the values of SPEI were calculated with the use of the FAO-56 Penman–Monteith equation, which is considered a superior method in reference to the Thornthwaite equation and recommended by Begueria et al. [29] as more robust for long-term climatological analysis. In addition, the method is based on longer time scales of 12, 24 and 30 months when presenting shortages of water. The study presented in this paper used the latest available data, which enhances accurate detection of changes in drought events. The main aim of this study was to find out how extreme was the spring drought of 2020, but it also checked the occurrence of the trend. By estimating drought events over longer time scales, this research took into account a lag in drought events, which is more visible in the long term and has a lower frequency. Short time scales are mainly related to soil water content; medium time scales are related to reservoir storages and discharge in the medium course of the rivers, and longer time scales are related to groundwater storage [25]. The SPEI 3 and SPEI 6, which use shorter scales, are masked by months with very high precipitation, such as February 2020. This wet month did not improve the hydrological drought situation but led to higher values of the SPEI. The SPEI value for the time scale of 12 months and more reflects well the occurrence of hydrological droughts with a substantial decrease in water resources, which may develop over a long period of time.

In order to enhance the results received through the SPEI, this research also studied combined satellite and observed data. The soil moisture contents were estimated based on data from the NASA Goddard Earth Sciences Data and Information Services Center (GES DISC) for several soil depth levels over the past 20 years: from January 2000 to May 2020.

## 2. Materials and Methods

In this paper, the SPEI and the soil moisture contents, based on gridded data from NASA, were used to illustrate scarcity of water in the landscape. The study area covers a belt of lowlands in central Poland with a longitude between 14.875° E and 23.625° E and latitude from 51.375° N to 53.375° N. In this area, only 11 meteorological stations are located, so data from another three sites situated close to the study area were added to enrich the data coverage (Figure 2).

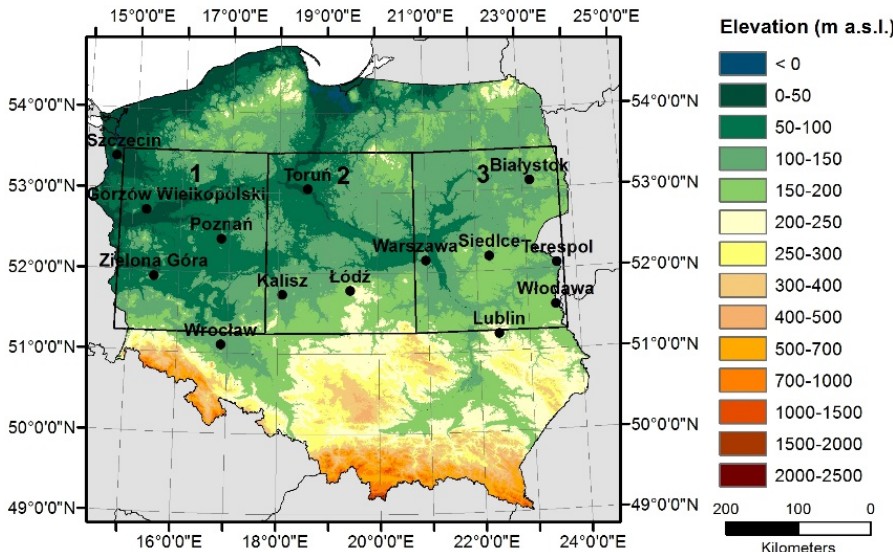

**Figure 2.** The study area covering the belt of lowlands in central Poland. Station data were provided by the Institute of Meteorology and Water Management—State Research Institute (IMGW-PIB). Data from the NASA Goddard Earth Sciences Data and Information Services Center (GES DISC) were divided into three equal subareas that were numbered, from west to east, as 1–3.

Data for the SPEI were obtained from 14 stations located in the belt of lowlands in central Poland (Figure 2), covering the interval from 1971 to the end of May 2020. These data sets were provided by the Institute of Meteorology and Water Management—State Research Institute (IMGW-PIB). Calculation of the SPEI was done in two steps. First, daily reference crop evapotranspiration ($ET_0$) in the R Evapotranspiration package was estimated [30,31], based on sub-daily and daily data. The data include: minimum and maximum temperature (in °C), minimum and maximum relative humidity (in %), insolation hours (available for eight stations: Białystok, Gorzów Wielkopolski, Kalisz, Poznań, Szczecin, Terespol, Włodawa and Zielona Góra) or cloudiness (available for the remaining six stations: Lublin, Łódź, Siedlce, Toruń, Warszawa and Wrocław), average wind speed (in m s$^{-1}$) and constants required for calculation of the Penman–Monteith FAO-56 formulation, including elevation (in m a.s.l.), latitude in radians and other constants.

In the second step, the monthly values of $ET_0$ and precipitation were calculated. Based on them, time series of the climatic water balance and then the SPEI were obtained in the R *SPEI* package [25]. The monthly data were then split into twelve series (one for each month) and independent PDFs (probability density functions) were fitted to each series (log-logistic distribution was used and parameter fitting was based on unbiased probability weighted moments). The SPEI was computed for three time scales: 12, 24 and 36 months. A period when the SPEI value continuously stayed below the threshold of −0.5 was defined as a duration of drought event, and its severity was defined as a cumulative deficit below the threshold of −0.5 [10].

Classification of severity of drought events, using SPEI values, follows Somorowska [14]:

- Moderate drought: $-1.0 \geq SPEI > -1.5$;
- Severe drought: $-1.5 \geq SPEI > -2.0$;
- Extreme drought: $-2.0 \geq SPEI$.

Drought periods with SPEI values continuously staying below −0.5 and lasting for at least five consecutive months, with a minimum value below −1.5, were identified in this study as severe droughts.

Additionally, the values of SPEI were examined for monthly series from January to May. It was checked if the values for these months of 2020 were below −2.0 (extreme drought) and if the value



for a particular month was the minimum in the whole data series. This aimed at examination of the severity of the 2020 spring drought in relation to other drought events.

Trend detection in series of $ET_0$ and the precipitation total for three stations (Poznań, Kalisz and Warszawa) and SPEI data was carried out with the use of the non-parametric Mann–Kendall test in the R *Trend* package [32], for two statistically significance levels: 0.05 and 0.1.

The soil moisture contents were estimated based on gridded data from the NASA Goddard Earth Sciences Data and Information Services Center [33]. The GLDAS-2.1 (Global Land Data Assimilation System) project uses a combination of the Noah land surface model and observation data made up of satellite and ground-based data and covering the period from January 2000 to May 2020. In this study, monthly data with a spatial resolution of 0.25 degrees were used. Outputs from GLDAS-2.1 include a wide variety of data, including the mean air temperature, precipitation total and soil moisture products in four layers: 0–10 cm, 10–40 cm, 40–100 cm and 100–200 cm.

The GLDAS-2.1 data for the belt of lowlands in central Poland with a longitude between 14.875° E and 23.625° E and latitude from 51.375° N to 53.375° N were divided into three equal subareas that were numbered, from west to east, as 1, 2 and 3 (Figure 2). Each subarea has a dimension of 12 columns and 9 rows. Gridded GLDAS-2.1 data (mean air temperature and precipitation total) were compared with observed data from the meteorological stations (data for a grid box and station data located in this grid box). For this purpose, one station for every subarea was chosen: data from Poznań for the first (west) area; data from Kalisz for the second (middle) area and data from Warszawa for the third (east) area. Data correlation for temperature was very high, with the coefficient of determination ($R^2$) exceeding 0.99 in all three cases; for precipitation, correlation was also quite good, with an $R^2$ above 0.73 for Poznań (first area) and Warszawa (third area) and above 0.71 for Kalisz (second area) (see Figure 3).

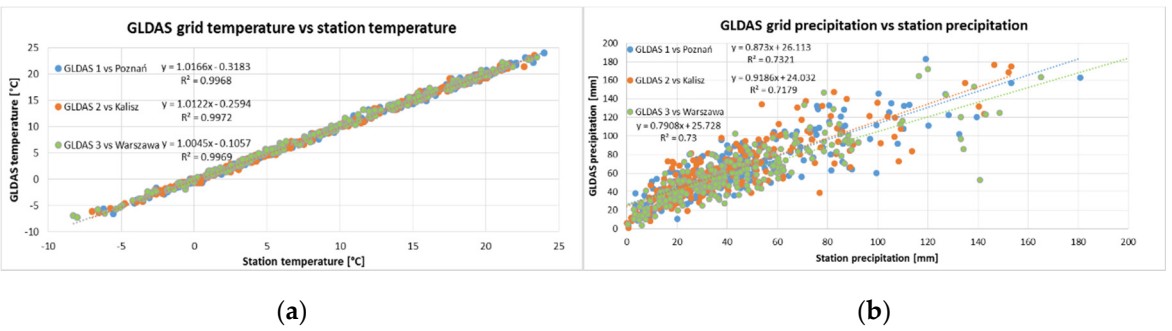

(**a**)  (**b**)

**Figure 3.** Comparison of monthly gridded GLDAS-2.1 (Global Land Data Assimilation System) data with observed data at meteorological stations from January 2000 to May 2020 for three studied areas: (**a**) for mean air temperature (in °C); (**b**) for precipitation total (in mm).

Changes in monthly soil moisture contents for several soil depth levels were examined for the aggregated area, i.e., mean values for cells from 12 columns and 9 rows for the period from January 2000 to May 2020.

## 3. Results

### 3.1. Severity of Drought Events Based on SPEI

Figure 4 presents the development of drought events for Poznań for three time scales of SPEI: SPEI 12, SPEI 24 and SPEI 30 in addition to the same indices based on the global drought monitor [34]. The SPEI indices for all three time scales for gridded data from January 1950 to May 2020 reveal that during the last drought event the drop was the highest for the whole period: SPEI 24 reached a minimum of −2.62 in January 2020; SPEI 30: −2.42 in April 2020; in both cases these results were one month earlier than those found through calculations in this study. For SPEI 12, the minimum

value occurred in March 2019 (−2.38). Generally, the courses of the SPEI for all three time scales for Poznań station and the gridded data were similar; however, the values for the gridded data were more strongly negative. Figures showing the results for the SPEI 12, 24 and 30 for 13 other analyzed stations are presented in the Supplementary Materials (see Figures S1–S3). Based on SPEI results, identification of drought periods that continuously stayed below −0.5 and lasted for at least five months with a minimum value below −1.5, i.e., severe drought events, was carried out. For each station there were several periods meeting these criteria. Tables S1–S3 in the Supplementary Materials presented these results for three time scales of SPEI: 12, 24 and 30 months. Based on them, in Tables 1–3, characteristics of the 2020 drought in relation to other drought events are presented. For only one of the 14 stations analyzed (Bialystok) there was no severe drought (meeting these criteria) for all SPEI calculations in 2020. At other stations, many characteristics of the 2020 drought were the most extreme for all SPEI time scales. For gridded data based on the global drought monitor, the SPEI for the grid of Białystok also showed very low values (see Figures S1–S3), in particular for SPEI 24 and SPEI 30—the values for the spring 2020 drought were the lowest there.

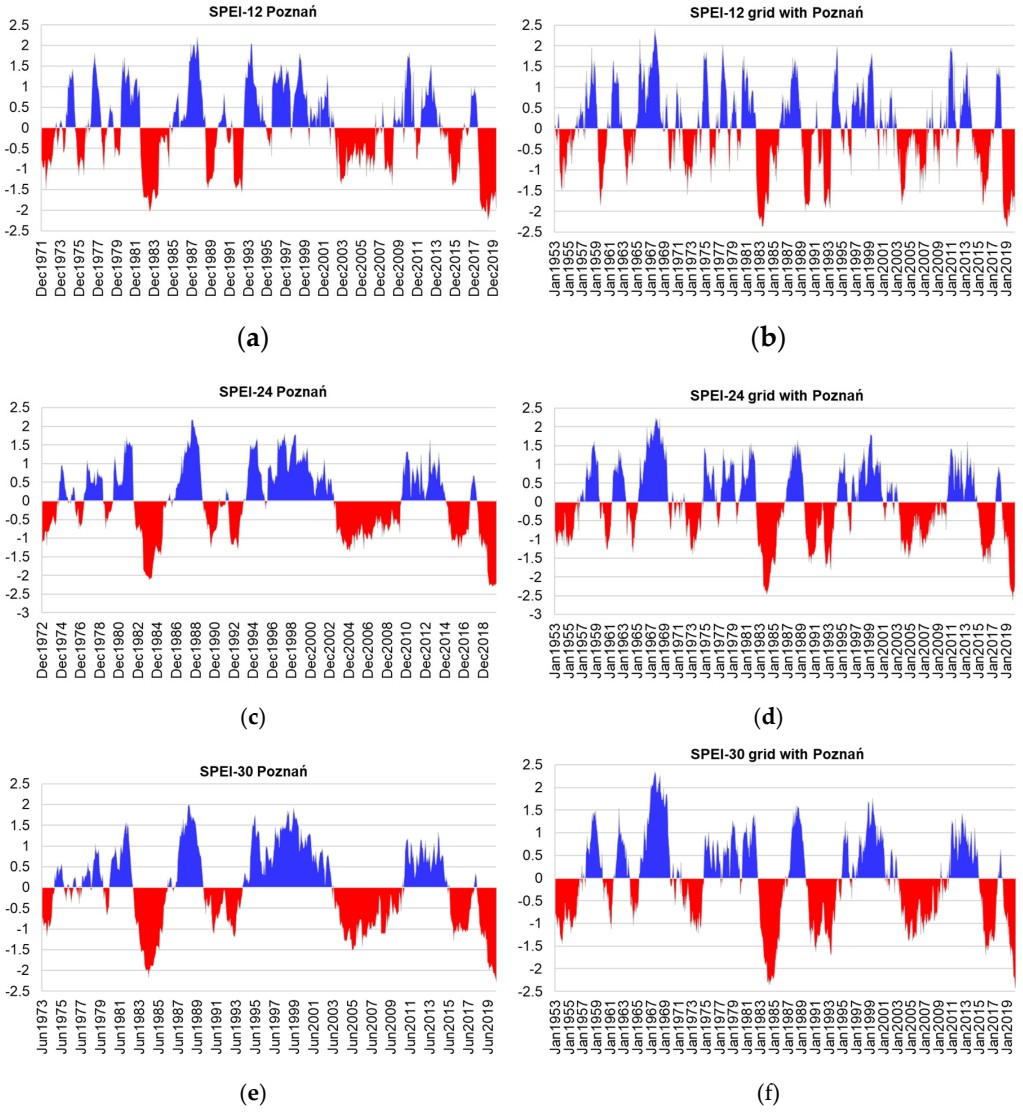

**Figure 4.** Values of SPEI 12 (Standardized Precipitation-Evapotranspiration Index for 12 months), SPEI 24 and SPEI 30 for Poznań from January 1971 to May 2020 based on station data (**a,c,e**) and gridded data for Poznań from January 1950 to May 2020 based on the SPEI global drought monitor (**b,d,e**).

### 3.1.1. SPEI 12

For SPEI 12, there were between two (Poznań) and eight (Lublin) periods meeting the above criteria (Table 1). At half of stations (seven of 14), the lowest minimum value for the period from January 1971 to May 2020 occurred during the last drought, lasting up to the end of May 2020. However, only in two cases these minimum values occurred in 2020. Despite the fact that only in four cases the spring drought of 2020 was the longest, in six cases the cumulative deficit below −0.5 was the most extreme. Our calculations were conducted until the end of May 2020, so these results could have been even higher if June and further months were considered, due to the occurrence of near-normal seasonal precipitation and of high temperatures during the summer of 2020, increasing evapotranspiration. Among earlier drought events, the drought of 2006 was the longest at four stations. Also in 2006, the most extreme value of SPEI 12 (−2.59 in Siedlce) was observed; whereas the most severe (with the lowest minimum) drought event occurred in 2015 (at two stations). For Białystok, the most severe droughts occurred in 1972 and 2000/2001 (see Figure S1 and Table S1).

**Table 1.** Characteristics of the 2020 spring drought event in relation to other drought periods in the time interval from January 1971 to May 2020, for which the value of SPEI 12 stayed below −0.5 and lasted at least five months, with a minimum value below −1.5 (definition of severe drought). * means that the drought was observed until the end of May 2020, so it may have lasted longer.

| Station | Duration of Drought Event (Months)/Ranking | Duration of Drought Events Max–Min/Number | Cumulative Deficit Below −0.5/Ranking | Minimum Value/Ranking | Data of Minimum Value Occurrence |
|---|---|---|---|---|---|
| Szczecin | 23 */1 | 23–16/5 | −36.26/1 | −2.30/1 | 19 April |
| Toruń | 21 */3 | 13–24/6 | −31.91/3 | −1.98/2 | 18 November |
| Białystok | - | - | - | - | - |
| Gorzów Wielkopolski | 18/4 | 23–13/6 | −26.96/2 | −2.13/2 | 19 April |
| Poznań | 23 */2 | 24–23/2 | −42.30/1 | −2.23/1 | 19 June |
| Warszawa | 24 */2 | 26–5/7 | −39.60/1 | −2.10/1 | 20 May |
| Siedlce | 12 */4 | 21–8/7 | −15.32/4 | −1.66/5 | 20 January |
| Terespol | 24 */1 (2) | 24–8/7 | −34.55/1 | −1.85/3 | 19 December |
| Zielona Góra | 23 */2 | 47–15/5 | −38.24/2 | −2.18/1 | 19 April |
| Wrocław | 26 */1 | 26–10/5 | −43.04/1 | −2.24/1 | 19 July |
| Kalisz | 23 */2 (2) | 40–12/7 | −35.10/2 | −1.77/3 | 19 June |
| Łódź | 21 */4 | 31–14/5 | −31.78/3 | −2.16/1 | 20 March |
| Lublin | 23 */1 | 23–6/8 | −36.18/1 | −2.32/1 | 19 April |
| Włodawa | 21 */2 | 25–8/8 | −29.52/2 | −1.93/7 | 20 January |

### 3.1.2. SPEI 24

Table 2 presents outcomes for drought events based on results of SPEI 24. In this case, for eight stations the lowest minimum value occurred during the 2020 spring drought and only in one case it was in December 2019 (Poznań); the other minimums were observed in 2020 (mainly in April). For one station (Wrocław), despite the shorter duration of the drought event (25 months; third longest), its cumulative deficit below −0.5 and minimum value were the lowest from all three drought events meeting the criteria (see Table S2). At Włodawa station, the most extreme value of SPEI 24 (−2.73 in September 2003) occurred during a drought lasting four years and covering two very dry periods in 2003 and 2006. Additionally, this period was the most severe for six stations, mainly due to the longest duration and the most extreme cumulative deficit below −0.5 (see Figure S2 and Table S2).

**Table 2.** Characteristics of the 2020 spring drought event in relation to other drought periods in the time interval from January 1971 to May 2020, for which SPEI 24 value stayed below −0.5 and lasted at least five months, with a minimum value below −1.5 (definition of severe drought). Symbol * means that the drought was observed until the end of May 2020, so it may have lasted longer.

| Station | Duration of Drought Event (Months)/Ranking | Duration of Drought Events Max–Min/Number | Cumulative Deficit Below −0.5/Ranking | Minimum Value/Ranking | Data of Minimum Value Occurrence |
|---|---|---|---|---|---|
| Szczecin | 12 */6 | 45–12/6 | −23.01/4 | −2.11/1 | 20 April |
| Toruń | 12 */6 | 29–12/6 | −21.20/5 | −2.14/2 | 20 April |
| Białystok | - | - | - | - | - |
| Gorzów Wielkopolski | 12 */4 | 59–12/4 | −19.78/4 | −1.92/2 | 20 May |
| Poznań | 23 */2 | 35–23/2 | −38.67/2 | −2.29/1 | 19 December |
| Warszawa | 15 */4 | 25–14/5 | −28.41/3 | −2.49/1 | 20 February |
| Siedlce | 12 */4 | 64–12/4 | −16.66/3 | −1.78/3 | 20 April |
| Terespol | 16 */3 | 52–15/4 | −27.05/3 | −2.27/1 | 20 April |
| Zielona Góra | 20 */4 | 54–13/5 | -28.95/3 | −2.12/1 | 20 January |
| Wrocław | 25 */3 | 38–25/3 | −41.18/1 | −2.34/1 | 20 April |
| Kalisz | 20 */5 | 41–20/5 | −27.72/4 | −2.05/1 | 20 April |
| Łódź | 12 */4 | 39–12/4 | −19.34/4 | −1.99/2 | 20 April |
| Lublin | 23 */2 | 33–12/4 | −37.95/2 | −2.39/1 | 20 April |
| Włodawa | 16 */2 | 48–6/6 | −23.89/2 | −2.03/2 | 19 December |

### 3.1.3. SPEI 30

For SPEI 30, the drought event recorded until the end of May 2020 for all stations (13 from 14) was not the longest one in the analyzed period for any station. However, for all stations, the minimum values of SPEI 30 were noted in April and May 2020, and for nine stations these values were the most extreme (Table 3). In April 2020 in Warszawa, SPEI 30 dropped to the extreme value of −2.61, while during the drought event beginning in 2015, this value was −2.03 in September 2016. As for SPEI 24, the most extreme value of SPEI 30 also occurred at Włodawa station (−2.67 in March 2004). The drought as a consequence of the dry years of 2003 and 2006 had the longest duration and the most extreme cumulative deficit below −0.5 for eight others stations (see Figure S3 and Table S3).

### 3.1.4. Extreme Values of the SPEI 12, 24 and 30 in 2020

The severity of the 2020 spring drought increases with the scale in SPEI. The SPEI 12 revealed extreme drought for months from January to May of 2020 (values below −2.0) only at a few stations. Additionally, values for particular months were not the most extreme for the whole analyzed period from January 1971 to May 2020 (Table 4). According to results of the SPEI 24, more severe conditions occurred at eight stations in January, February and March and the index reached a value below −2.0; in April it was reached at nine stations, and in May at five stations. Additionally, at eight stations, conditions for all months from January to May were the most severe. In April 2020, the driest conditions were noted for 12 of the 14 analyzed stations. SPEI 30 values below −2.0 occurred mainly in April (10 times) and May (11 times) and at half of all (seven of 14) stations; these values were the most extreme for the months from January to May.

**Table 3.** Characteristics of the 2020 spring drought event in relation to other drought periods in the time interval from January 1971 to May 2020, for which SPEI 30 value stayed below −0.5 and lasted at least five months, with a minimum value below −1.5 (definition of severe drought). Symbol * means that the drought was observed until the end of May 2020, so it may have lasted longer.

| Station | Duration of Drought Event (Months)/Ranking | Duration of Drought Events Max–Min/Number | Cumulative Deficit Below −0.5/Ranking | Minimum Value/Ranking | Data of Minimum Value Occurrence |
|---|---|---|---|---|---|
| Szczecin | 11 */5 | 39−11/5 | -14.75/5 | −2.13/1 | 20 May |
| Toruń | 10 */6 | 27−10/6 | -12.87/6 | −2.13/2 | 20 April |
| Białystok | - | - | - | - | - |
| Gorzów Wielkopolski | 7 */4 | 60−7/4 | −9.37/4 | −1.86/2 | 20 May |
| Poznań | 22 */3 | 49−22/3 | −35.13/3 | −2.26/1 | 20 May-20 |
| Warszawa | 12 */6 | 36−12/6 | −22.87/6 | −2.61/1 | 20 April |
| Siedlce | 11 */3 | 63−11/3 | −11.86/3 | −1.86/2 | 20 April |
| Terespol | 14 */3 | 59−14/3 | −22.26/3 | −2.20/1 | 20 April |
| Zielona Góra | 14 */3 | 49−14/3 | −20.60/3 | −2.06/1 | 20 May |
| Wrocław | 24 */4 | 66−24/4 | −38.15/2 | −2.44/1 | 20 April |
| Kalisz | 14 */4 | 53−14/4 | −20.73/4 | −2.13/1 | 20 April |
| Łódź | 10 */5 | 47−10/5 | −12.47/5 | −2.13/1 | 20 April |
| Lublin | 22 */2 | 32−22/2 | −35.25/2 | −2.51/1 | 20 April |
| Włodawa | 14 */3 | 49−14/3 | −20.47/3 | −2.14/2 | 20 April |

**Table 4.** Extreme values of the SPEI 12, 24 and 30 for the months from January to May 2020.

| Station | SPEI 12 | | SPEI 24 | | SPEI 30 | |
|---|---|---|---|---|---|---|
| | Value ≤ −2.0 | Record | Value ≤ −2.0 | Record | Value ≤ −2.0 | Record |
| Szczecin | - | - | J F M A M | J F M A M | _ _ _ _ M | J F M A M |
| Toruń | - | - | J F M A _ | J _ M A _ | _ _ _ A M | _ _ _ A M |
| Białystok | - | - | - | - | - | - |
| Gorzów Wielkopolski | - | - | - | J _ _ A _ | - | - |
| Poznań | - | _ _ _ _ M | J F M A M | J F M A M | J F M A M | J F M A M |
| Warszawa | J F M _ M | J F M A M | J F M A M | J F M A M | _ _ M A M | J F M A M |
| Siedlce | - | - | - | - | - | _ _ _ A M |
| Terespol | - | J F M A M | J F M A _ | _ _ M A M | _ _ M A M | J F M A M |
| Zielona Góra | - | - | J F M A M | J F M A M | _ _ _ A M | J F M A M |
| Wrocław | - | _ _ _ A M | J F M A M | J F M A M | J F M A M | J F M A M |
| Kalisz | - | _ _ _ A _ | _ _ _ A _ | J F M A M | _ _ _ A M | J F M A M |
| Łódź | J F M A _ | J F M A _ | - | J F M A M | _ _ _ A M | _ _ M A M |
| Lublin | - | - | J F M A _ | J F M A M | J _ M A M | J _ M A M |
| Włodawa | - | - | - | _ _ _ A _ | _ _ _ A M | - |

### 3.1.5. Trend Detection in Annual $ET_0$ and Precipitation Total for Three Stations: Poznań, Kalisz and Warszawa

At all three analyzed stations, the Mann–Kendall test revealed statistically significant (below 0.05) increasing trends for reference crop evapotranspiration ($ET_0$). The precipitation totals for Poznań and Warszawa show an increasing tendency and a decreasing one for Kalisz, but all three changes were statistically insignificant (Figure 5). Maximum values of annual $ET_0$ were recorded during the last severe droughts: 2015, 2018 and 2019, and also meet one of the minimum values of precipitation:

for Kalisz the maximum $ET_0$ occurred in 2015 (861 mm) with a minimum precipitation total of 259 mm; in Poznań in 2018 (931 mm) with a precipitation of 379 mm; and in Warszawa in 2019 (947 mm) with a precipitation of 390 mm.

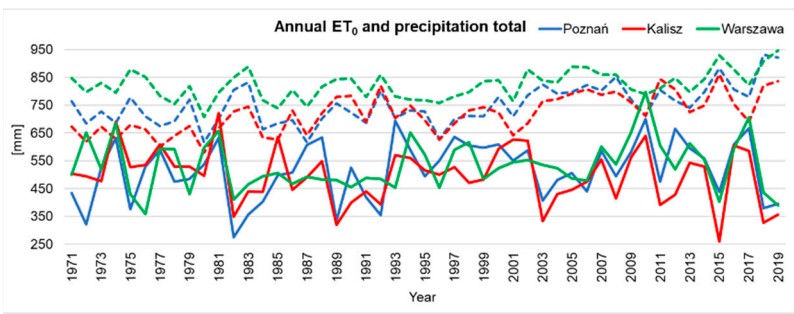

**Figure 5.** Annual reference crop evapotranspiration ($ET_0$) (dashed line) and precipitation total (solid line) for three station: Poznań, Kalisz and Warszawa for the period of 1971–2019.

### 3.1.6. Trend Detection in SPEI 12, 24 and 30

At most analyzed stations, the Mann–Kendall test detected decreasing trends for all scales of SPEI. For SPEI 12, a statistically significant (at the level of 0.05) decreasing trend was detected at 10 of the 14 stations. For an additional two stations, Warszawa and Siedlce, the decreasing trend was weaker but still statistically significant at the level of 0.1, and for two remaining stations (Szczecin and Białystok), the trend was insignificant.

For SPEI 24, at 12 stations, a statistically significant (at the level of 0.05) decreasing trend was detected and at one additional station (Siedlce), it was detected at the level of 0.1, while at Szczecin no change was detected. Similar results can be reported for SPEI 30: 11 statistically significant (at the level of 0.05) decreasing trends, two weaker (at the level of 0.1) trends at Warszawa and Siedlce and no statistically significant trend for Szczecin.

### 3.2. Soil Moisture Changes Based on GLDAS-2.1 Data

Figure 6 presents the percentage decrease in monthly soil moisture contents in 2020 relative to the monthly mean (2000–2019) soil moisture contents at two depth ranges, 40 to 100 cm and 100 to 200 cm, in April and May. In eastern Poland for the shallower layer for April 2020 this decrease dropped by up to 25% (Figure 6a). The soil moisture content ranged for this layer from 40 to 220 kg m$^{-2}$. A decrease for May 2020 at the depth range from 100 to 200 cm reached up to 12% in the eastern part of the country (Figure 6d). Figure 6c,d depicts a deep layer with soil moisture contents between 180 and 320 kg m$^{-2}$. Additionally, it is worth pointing out that the 2000–2019 mean includes very dry years: 2003, 2006, 2015, 2018 and 2019. A decrease in monthly soil moisture contents in April and May 2020 was the highest in the eastern part of the studied area. This part has a greater annual precipitation (above 600 mm) than the central and western parts (below 600 mm). Furthermore, the amount of snow cover was larger and the annual temperature was lower [15], so the monthly 2000–2019 mean values of soil moisture content were higher there: 157, 166 and 168 kg m$^{-2}$ for the first, second and third parts, respectively, for the depth range from 40 to 100 cm in April 2020; and 255, 270 and 268 kg m$^{-2}$, respectively, for the depth range from 100 to 200 cm in May. After precipitation in May 2020 (below normal in the western, near normal in the central and above normal in the eastern part of the study), moisture conditions in the shallower layer improved (40–100 cm, Figure 6c) and became worse in the deeper layer (100–200 cm, Figure 6d).

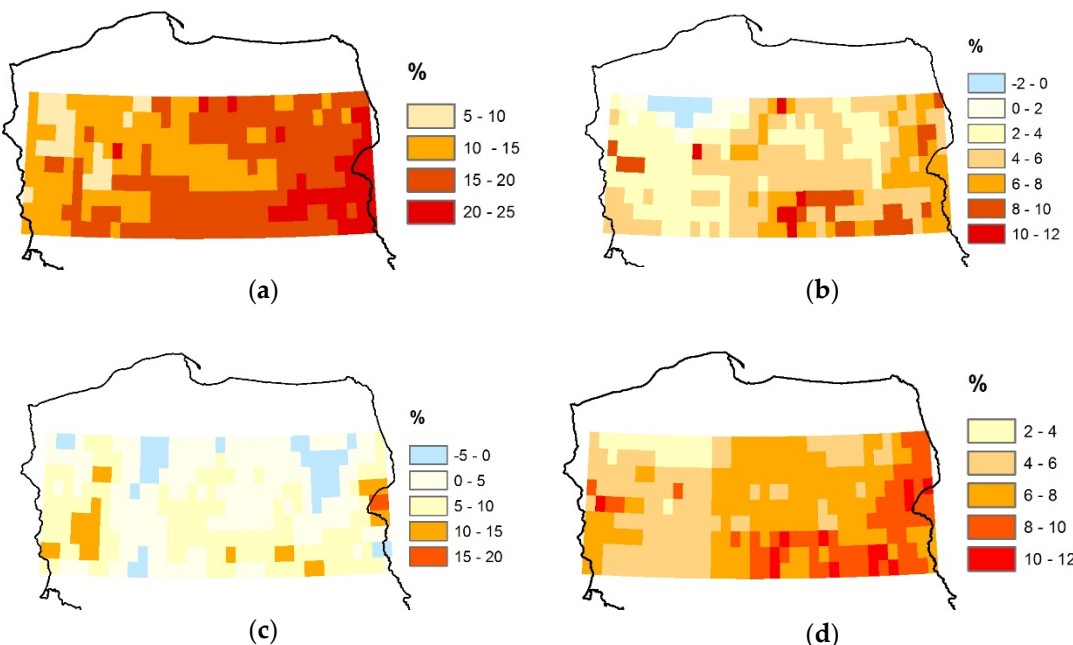

**Figure 6.** Percent decreases in monthly soil moisture contents in 2020 relative to the monthly 2000–2019 mean soil moisture contents for: (**a**) April at 40–100 cm depth; (**b**) April at 100–200 cm depth; (**c**) May at 40–100 cm depth; (**d**) May at 100–200 cm depth. Data are from the NASA Goddard Earth Sciences Data and Information Services Center (GES DISC).

For all depth layers and studied areas, a decreasing tendency in soil moisture contents during the period from January 2000 to May 2020 was detected. The most shallow layer of soil, at the depth range from 0 to 10 cm, is the most fragile as regards transpiration and loss of water, but soil moisture in this layer also replenishes quickly. For all three areas, the lowest values were visible for summer 2015 and then for 2006. One of the lowest maxima for winter precipitation was observed during 2020 (Figure 7).

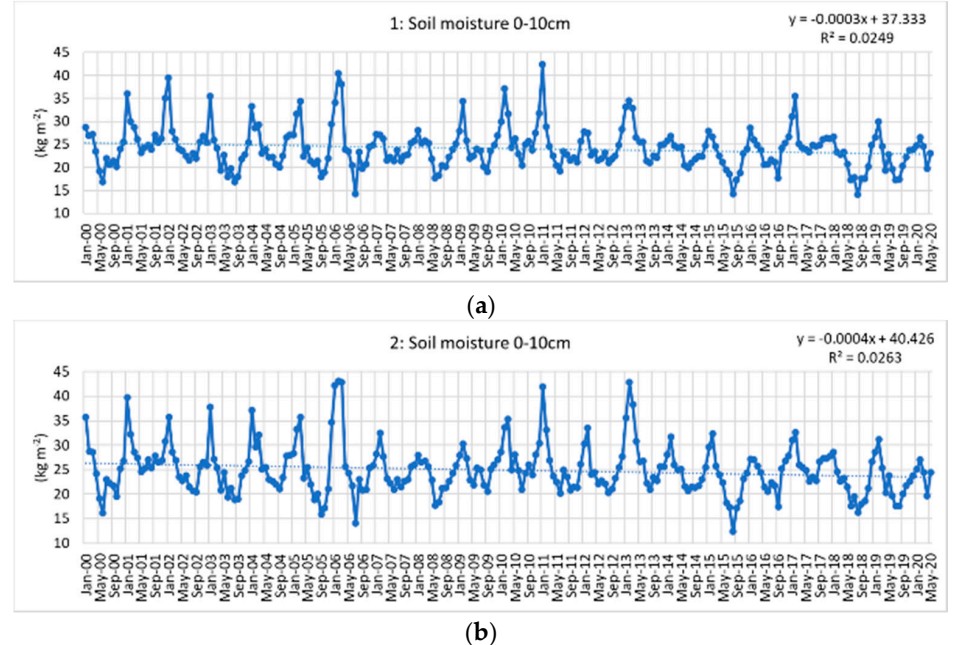

**Figure 7.** *Cont.*

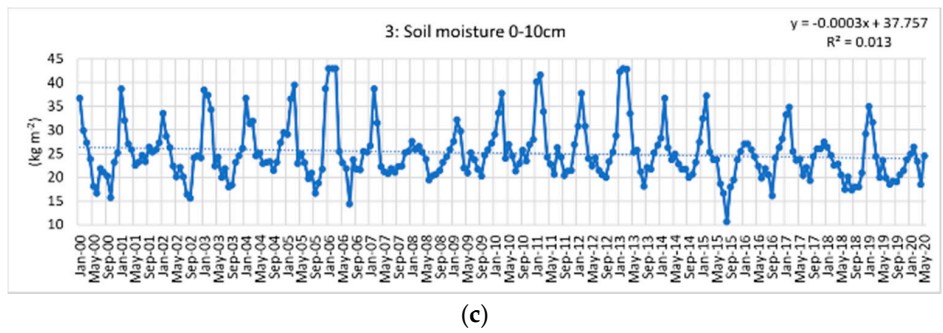

(c)

**Figure 7.** Monthly soil moisture contents in kg m$^{-2}$ during the period from January 2000 to May 2020 for a range of depths from 0 to 10 cm, for (**a**) the first; (**b**) the second; and (**c**) the third subareas.

Similar behavior can be observed for a deeper soil layer, from 10 to 40 cm below the ground surface (Figure 8). Especially for the first area, one can notice very small moisture recovery for the most recent winters, and the deepest summer excursions were in 2015 followed by 2018 and 2019. Only for the eastern (third) area, more moisture was recorded during winters, apart for winter 2020.

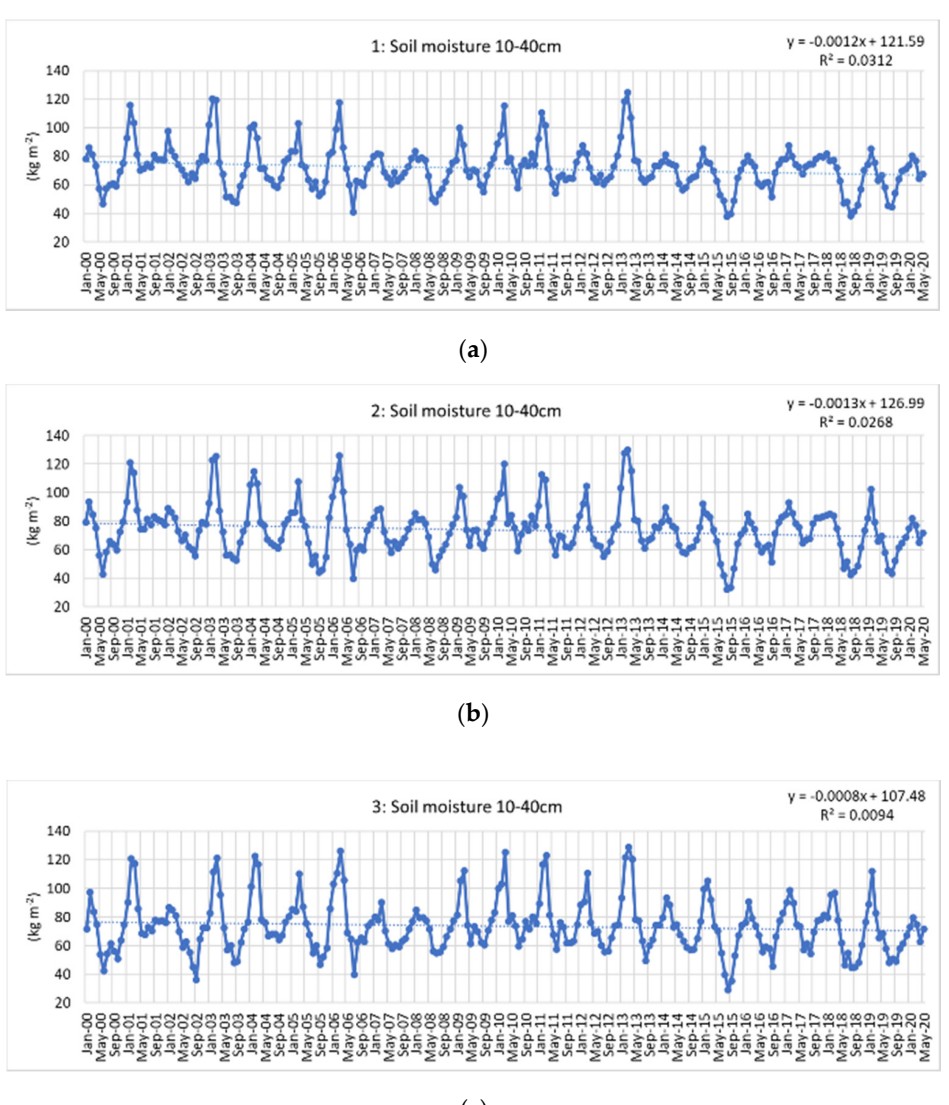

(a)

(b)

(c)

**Figure 8.** Monthly soil moisture contents in kg m$^{-2}$ during the period from January 2000 to May 2020 for a range of depths from 10 to 40 cm, for (**a**) the first; (**b**) the second; and (**c**) the third subareas.

For the soil layer from 40 to 100 cm (Figure 9) the deepest excursions were also visible for the summers of 2015 and 2018. For April 2019 and 2020, the soil moisture contents were the lowest for all studied periods.

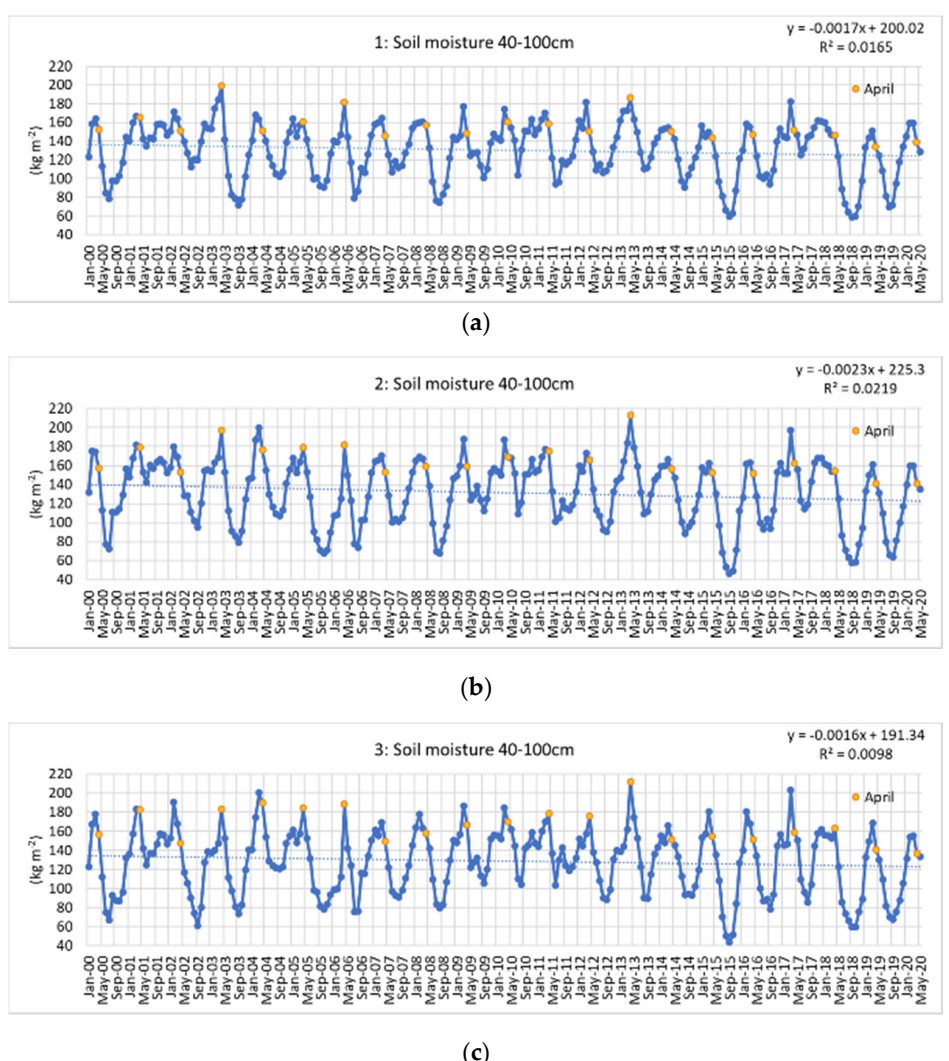

(a)

(b)

(c)

**Figure 9.** Monthly soil moisture contents in kg m$^{-2}$ during the period from January 2000 to May 2020 for a range of depths from 40 to 100 cm, for (**a**) the first; (**b**) the second; and (**c**) the third subareas.

The deepest soil layer, from 100 to 200 cm, is the least sensitive to fast changes at the soil surface (Figure 10). However, for the first area, shortages of water were quite well visible during dry years: 2015, 2018 and 2019. For all three areas, a very small maximum can be noticed for the winter of 2019. Additionally, values for May 2018, 2019 and 2020 were among the smallest in the time series.

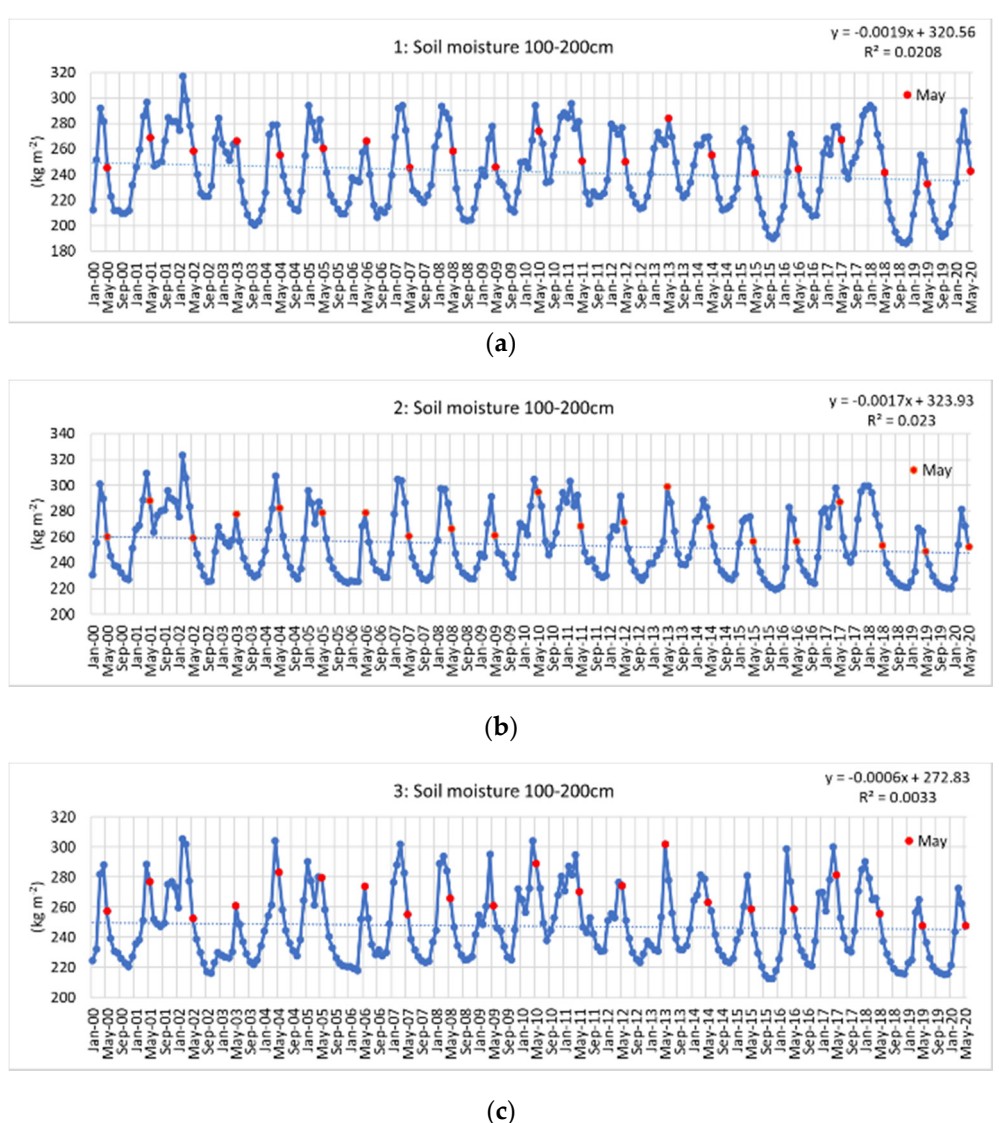

**Figure 10.** Monthly soil moisture contents in kg m$^{-2}$ during the period from January 2000 to May 2020 for a range of depths from 100 to 200 cm, for (**a**) the first; (**b**) the second; and (**c**) the third subareas.

## 4. Discussion

In Poland, seasonal precipitation during autumns and winters should improve the soil moisture conditions after warm or warm and dry springs and summers. If during a cold season precipitation totals are below or near normal, at the beginning of the next warm season, the moisture conditions could be insufficient and contribute to a drought. Since 2015 until the present, 2017 was the only year with a precipitation total considerably above the long-term mean. Additionally, the years 2015, 2018 and 2019 witnessed higher-than-normal air temperatures and heat waves, which exacerbated shortages of water. This situation also coincides with the occurrence of small (or even non-existent) snow cover in winter, leading to increasingly drier soil conditions.

In this research, it was shown that the 2020 spring drought was one of the most severe in the last five decades. It also presented decreasing trends of SPEI at most stations located in central Poland. Cumulative water shortages from year to year, despite very high precipitation in February 2020, led to the development of severe drought, as reflected in very low SPEI values.

The presented soil moisture contents for all studied depth layers from January 2000 to May 2020 indicate a decreasing tendency. This shift to drier soil conditions is a result of the occurrence of dry and warm years, but also higher temperatures during all seasons. Observed warming at a range of

scales, from global to regional [35–39], has manifested itself not only in higher temperatures, but also in a reduced amount of snowfall and snow cover and an increase to the liquid phase of precipitation in winter [40,41]. This was also the case in February 2020, which was rainy but snowless over much of Poland. Pińskwar et al. [24] examined changes in 1961–2017, showing precipitation increases for February, March, July, September and October and decreases for June, August, November and December. In addition, warm months, like April, June and August, exhibited a drying tendency. In April 2020, precipitation was exceptionally low, so that the water deficit strengthened [15].

Another issue is the temporal distribution of precipitation within a month. The monthly precipitation total can be close to or above the multi-year mean value, but its distribution is uneven, especially during warm months [42,43]. One could observe this especially in June 2020 at Warszawa station, where the total precipitation was 160 mm, while nearly 100 mm accumulated in just four days with a maximum of 41 mm. The spatial distribution of precipitation also plays an important role. Abundant rainfall occurred much less in central Poland than in the south of country, where mountainous regions dominate. The third part of June 2020 was very wet in the south of Poland, with up to a 24 h rainfall of 152 mm recorded in Jodłownik (Małopolskie Voivodship). This heavy rain led to a massive local increase of the water stage in the Stradomka River —by five meters in less than one day. In effect, a large part of the municipality of Łapanów (County of Limanowa) was inundated. On several gauges of the two main rivers in Poland, the Vistula and the Odra, the alarm stages were exceeded. For instance, on 21 June, the alarm stage was exceeded at 22 gauges in the Odra River basin, while on 23 June, the alarm stage was exceeded at 11 gauges in the Vistula River basin (data from IMGW-PIB).

For three analyzed stations, the worrying situation is that trends for reference crop evapotranspiration ($ET_0$) significantly increased and at the same time the precipitation total remained nearly unchanged (slightly increased or decreased). Additionally, as Javadian et al. demonstrated [44], actual evapotranspiration (AET) has significantly increased across global croplands (+14% ± 5%). Global analysis shows that the increasing trend in AET is also visible in Poland.

Cumulative precipitation shortages can cause a hydrological response, i.e., drought for a longer period. Even if months with higher precipitation may occur, the surplus can be insufficient to compensate for the accumulated deficit. Yet, they can mask drought when analysis is carried out for a shorter period.

The water resources of Poland (precipitation and river runoff, at the national and per capita scales) are low—lower than those of most countries in Europe [23]. Precipitation increases during the cold season (especially in February and March), when water does not stay in the landscape and the evaporation may be higher because of higher temperatures. During warmer months, prolonged dry spells (interrupted by intense precipitation events), high evapotranspiration and occurrence of heat waves may lead to the development of a multi-year drought. In this sense, the drought in the spring of 2020 in Poland should not be regarded as more of the same, or something extraordinary, but more as a harbinger of things to come. Therefore, due to an increasing risk of such events, and the rising severity of their consequences, studies dealing with the problem are of high importance.

## 5. Conclusions

Droughts are major weather-driven natural disasters that can occur everywhere, including in water-rich areas, due to occasional anomalies in climatic variables. They can last long and encompass large areas. Drought losses have significantly increased in recent years worldwide and for a range of reasons, including climatic and non-climatic factors [45]. In Europe, the humid north is likely to become even more humid, while the dry south is likely to become even drier [46]. As Poland is located in between these regions, its drought signal is more complex.

New and detailed knowledge shedding light on the problem of droughts and their specificity may contribute to a better adaptation and introduction of policy preparedness. Above all, the spring 2020 drought in Poland, which is now seen as an extraordinary event, as analyses indicate, does not stand out of the future norm and similar droughts will become a frequent occurrence. Moreover, the frequency and magnitude of drought events may increase not only in Poland, but in much of Europe

as well. Extreme droughts from the past, e.g., those recorded in 2003, 2010, 2015 and 2018–2019, will no longer be classified as extreme in Europe. Without effective climate change mitigation, large areas of Europe may be affected by soil moisture deficits [3,20,21].

Rain-fed agriculture is likely to face increasing water stress during dry years. Improving of water use efficiency (WUE) is one of the most significant challenges. Among the agrotechnical practices one can find subsoiling, which improves the permeability of soil water; it not only disturbs and breaks the plough layer, but also creates additional water storage [47]. Another practice that can improve WUE is mulching that reduces soil water evaporation. Especially organic mulching increases WUE by improving the physical properties of the top layer of the soil [48]. In addition, the strategy of reducing row spacing between crops can be followed, as it reduces evaporation; another strategy is the diversification of crop rotation in order to increase the resilience of the overall cropping system [49]. Another issue is the improving of micro-climates through the introduction of trees or shrubs with annual crops in the same area [49] or small ponds.

Having too little or too much water can become increasingly frequent problems in our future and coping with these problems will be one of the major challenges for society.

**Supplementary Materials:** The following Supplementary Materials are available online at http://www.mdpi.com/2073-4395/10/11/1646/s1: Figure S1: Values of SPEI 12 for 13 analyzed stations from January 1971 to May 2020 (left column) and for grid with stations from January 1950 to May 2020 based on SPEI Global Drought Monitor (right column). Figure S2: Values of SPEI 24 for 13 analyzed stations from January 1971 to May 2020 (left column) and for grid with stations from January 1950 to May 2020 based on SPEI Global Drought Monitor (right column). Figure S3: Values of SPEI 30 for 13 analyzed stations from January 1971 to May 2020 (left column) and for grid with stations from January 1950 to May 2020 based on SPEI Global Drought Monitor (right column). Table S1: The drought periods for which the value of SPEI 12 stayed below −0.5 and lasted at least five months with minimum value below −1.5 (definition of severe drought), based on results of SPEI 12. The most severe values are marked in bold. Table S2: The drought periods for which the value of SPEI 24 stayed below −0.5 and lasted at least five months with minimum value below −1.5 (definition of severe drought), based on results of SPEI 24. The most severe values are marked in bold. Table S3: The drought periods for which the value of SPEI 30 stayed below −0.5 and lasted at least five months with minimum value below −1.5 (definition of severe drought), based on results of SPEI 30. The most severe values are marked in bold.

**Author Contributions:** Conceptualization, I.P. and Z.W.K.; methodology, I.P.; software, I.P.; formal analysis, I.P.; writing—original draft preparation, I.P., Z.W.K. and A.C.; writing—review and editing, Z.W.K., A.C., I.P.; visualization, I.P. All authors have read and agreed to the published version of the manuscript.

**Funding:** This research received no external funding.

**Acknowledgments:** The four anonymous referees are acknowledged for their valuable comments.

**Conflicts of Interest:** The authors declare no conflict of interest.

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
