# Peer review of "Severe Drought in the Spring of 2020 in Poland—More of the Same?"

_agronomy, doi:10.3390/agronomy10111646_

Round 1

Reviewer 1 Report

I feel that the paper is comprehensive and well-structured. It concerns the important issue, especially for Polish agriculture. The introduction is long, however, well-prepared. The results are presented in the correct way, discussion and conclusions are in my opinion, justified.

Author Response

The authors thank for kind review.

Reviewer 2 Report

The paper manuscript entitled "Severe drought in the spring of 2020 in Poland. More of the same?

The authors tried to present a study for drought investigation in Poland based on the SPEI index.

General comments:

The work is good, but it still needs some improvements and corrections.

Specific comments:

My specific comments I have written on the manuscript pages and downloaded to the Editor, who should send it to the author.

Besides, my specific comments are as follows:

Page 1, Line 9: How you know there is a warm winter 2020, while you are in Sep. 2020?

Page 1, Line 10: It is better here to mention the study time period.

Page 1, Line 16: It is much better to use (the study) instead of (we) everywhere in your paper.

Page 1, Line 21: the author should add the main conclusion extracted from his study at the end of the (Abstract) selection.

Page 1, Line 22: Add (drought) as a keyword.

Page 3, Line 95: Explain which Metop satellite’s data you have used; A or B or C?

Page 4, Line 144: Why 2020? You have not the full data of all months of 2020 to be able to decide whether it is a drought year, or not?

Page 5, Line 195: please explain the statement colored by yellow.

Page 5, Line 201: Correct (0.1) to be (0.01).

Page 6, Line 220: The two figures are missing the X & Y axises titles.

Page 9, Line 305: Is there any statistical relation has been calculated between the drought indicators and the geographical locations (lat., long.) of the stations? Why you did not investigate it?

Page 10, Line 328: How can you do compare for study cases with different depth?

Page 10, Line 331: Is that happened in all the study years?

Page 13, Line 356: I suggest separating the discussion and conclusions into two sections; (4. Discussion) and (5. Conclusions).

Page 13, Line 359: Rewrite the sentence. It is a vague one.

Page 13, Line 369: Indicate the depth.

Page 13, Line 380: What about the spatial distribution?

Author Response

The authors thank for valuable comments.

Specific comments:

My specific comments I have written on the manuscript pages and downloaded to the Editor, who should send it to the author.

Besides, my specific comments are as follows:

Page 1, Line 9: How you know there is a warm winter 2020, while you are in Sep. 2020?

It has been changed to 2019/20.

Page 1, Line 10: It is better here to mention the study time period.

It has been changed.

Page 1, Line 16: It is much better to use (the study) instead of (we) everywhere in your paper.

It has been changed.

Page 1, Line 21: the author should add the main conclusion extracted from his study at the end of the (Abstract) selection.

There is a limit of 200 words.

Page 1, Line 22: Add (drought) as a keyword.

There is a limit of 3 keywords.

Page 3, Line 95: Explain which Metop satellite’s data you have used; A or B or C?

This illustration stems from a web page of the Polish Institute of Meteorology and Water Management (IMGW). Details of the data set (whether A or B or C) are not clear, yet, we trust that - with appropriate reference to IMGW – this map is meaningful and serves a useful purpose.

Page 4, Line 144: Why 2020? You have not the full data of all months of 2020 to be able to decide whether it is a drought year, or not?

It has been changed to spring 2020.

Page 5, Line 195: please explain the statement colored by yellow.

It has been explained.

Page 5, Line 201: Correct (0.1) to be (0.01).

It is correct. We take 90% (or 0.1), i.e. weaker significance level, rather than 99% (or 0.01).

Page 6, Line 220: The two figures are missing the X & Y axises titles.

It has been changed.

Page 9, Line 305: Is there any statistical relation has been calculated between the drought indicators and the geographical locations (lat., long.) of the stations? Why you did not investigate it?

We did not calculate any statistical relation between the drought indices and the geographical locations. Drought is a phenomenon that occurred on greater area in the same time.  Based on our calculations, SPEI index showed very low values at nearly all studied area. Only at BiaĹ‚ystok station the value of SPEI were higher. SPEI values for one point are different for gridded data, where for grid with BiaĹ‚ystok SPEI also shows very low values (see Supplementary Materials), in particular for SPEI 24 and SPEI 30 – the values for spring 2020 drought are the lowest.

Page 10, Line 328: How can you do compare for study cases with different depth?

The section has been changed. We added also figures for April (100-200cm) and for May (40-100 cm).

Page 10, Line 331: Is that happened in all the study years?

Decreasing tendency was calculated based on linear regression for the complete period from January 2000 to May 2020.

Page 13, Line 356: I suggest separating the discussion and conclusions into two sections; (4. Discussion) and (5. Conclusions).

It has been changed.

Page 13, Line 359: Rewrite the sentence. It is a vague one.

It has been changed.

Page 13, Line 369: Indicate the depth.

It has been changed.

Page 13, Line 380: What about the spatial distribution?

Text was added.

Reviewer 3 Report

Very important and interesting news for agricultural practice. The article is very interesting.
Comments on the article:

  1. Please modify the title of the article, remove the year 2020 from it
    2. SPEI keywords do not match the keywords
    3. Present the agrotechnical recommendations in the summary of the work. Utilitarian conclusion on how to proceed in such conditions. What a farmer (agriculture) can do to protect the plants from the moisture deficiency in the soil
    4. Verify the literature more carefully

Final conclusion

In the form presented, the paper brings a lot of new information. It does not have to be completely rewritten and corrected. In summary, it should be stated that the work in the presented scope is interesting and suitable for printing after taking into account the comments of the Reviewer.

Author Response

The authors thank for valuable comments.

Comments on the article:

  1. Please modify the title of the article, remove the year 2020 from it

Why? It is important to mark the “spring of 2020”, because it was quite special.

  1. SPEI keywords do not match the keywords

It has been changed to Standardised Precipitation-Evapotranspiration Index.

  1. Present the agrotechnical recommendations in the summary of the work. Utilitarian conclusion on how to proceed in such conditions. What a farmer (agriculture) can do to protect the plants from the moisture deficiency in the soil

The text was added:

Rain-fed agriculture is likely to face an increasing water stress during dry years. Improving of water use efficiency (WUE) is one of the most significant challenges. Among the agrotechnical practices one can find subsoiling, which improves the permeability of soil water, it not only disturbs and breaks the plough layer, but also creates additional water storage [47]. Another practice which can improve WUE is mulching that reduces soil water evaporation. Especially organic mulching increases WUE by improving the physical properties of the top layer of soil [48]. Also the strategy of reducing row spacing between crops can be followed, because of evaporation reduction and diversification of crop rotation in order to increase resilience of the overall cropping system [49]. Another issue is the improving of micro-climate by introduction of trees or shrubs with annual crops on the same area [49] or small ponds.

  1. Mohanty, M.; Bandyopadhyay, K.K.; Painuli, D.K.; Ghosh, P.K.; Misra, A.K.; Hai, K.M. Water transmission characteristics of a Vertisol and water use efficiency of rainfed soybean (Glycine max (L.) Merr.) under subsoiling and manuring. Soil And Till. Res., 2007. 93, 2, p. 420-428.
  2. Lordan, J.; Pascual, M.; Villar, J.M.; Fonseca, F.; Papió, J.; Montilla, V.; Rufat, J. Use of organic mulch to enhance water-use efficiency and peach production under limiting soil conditions in a three-year-old orchard. Spanish Journal of Agricultural Research 2015, 13, 4, e0904, http://dx.doi.org/10.5424/sjar/2015134-6694
  3. Hatfield J.L.; Dold, Ch. Water-Use Efficiency: Advances and challenges in a changing climate. Front. Plant Sci., 2019, https://doi.org/10.3389/fpls.2019.00103
  4. Verify the literature more carefully

The literature has been carefully verified. Several further references of relevance have been included.

Reviewer 4 Report

The paper titled “Severe drought in the spring of 2020 in Poland. More of the same?” investigated a severe drought in Poland. Generally, the direction of the paper is interesting. The authors' findings can be used in upcoming studies. However, there are some concerns about the paper. I encourage the authors to compare their SPEI results with some popular SPEI products to make sure about their calculations. Furthermore, it would be better to compare some other trends with the SPEI trends which I mentioned in the comments. Therefore, I am recommending publication only after the substantial rewriting of the proposed paper is complete. In addition to this, more specific comments are given below. Thank you for the opportunity to review this paper.

Line 9-11: It would be better to add something about the reason for drought analysis. Maybe water stress or something else.

Line 71: the sentence “over several years now, winters in Poland have 70 been frequently warmer than a long-term average.” needs a reference. There are some other sentences in the introduction that needs references. Please add relevant references if needed.

Line 74-90: There is no reference here!

Line 140-141: What is the merit of this novelty?

Line 144: What do you mean in the sentence “but we also learned the occurrence of trend”?

Line 171: Why didn’t you obtain SPEI from the available SPEI products?

One of them: https://spei.csic.es/database.html

Line 203: It would be better to check out the recent version of GLDAS which is GLDAS 2.2. If it’s possible just compare a sample of data

Line 243: Please increase the quality of your figures. You also need to introduce each plot separately in the caption.

Line 307: Please briefly compare the trends of actual ET, potential ET, and precipitation with your SPEI result at least for a few of your stations. Use the application of the paper below:

https://www.mdpi.com/2072-4292/12/7/1221

Line 327: Why there is a huge drop in soil moisture in the Southeastern side? Please discuss it.

Line 336: Please reproduce the plots. They don’t have publication quality.

Line 339: Why there was a moisture recovery for last winters?

Line 356: I would like to read the application of your research in the big picture, not just for your case study. Please add a paragraph for connecting your research to global drought studies.

Author Response

The authors thank for valuable comments.

Line 9-11: It would be better to add something about the reason for drought analysis. Maybe water stress or something else.

The text was added in Introduction:

Due to low mean annual precipitation (national average 624 mm) [22] and low water resources (the mean about 60 billion m3, in dry years much less: below 40 billion m3), the per capita water availability in Poland is far below the European average. It is only possible to store approximately 6.5% of annual river flow in reservoirs in Poland [23], hence there would be no water for massive agricultural irrigations in the country that may be needed in the warming climate. The water stress is on the rise in Poland, hence drought analysis is of much relevance and interest in the nation.

  1. Wibig, J., Jakusik E., eds. Warunki klimatyczne i oceanograficzne w Polsce i na BaĹ‚tyku poĹ‚udniowym – spodziewane zmiany i wytyczne do opracowania strategii adaptacyjnych w gospodarce krajowej (Climatic and oceanographic conditions in Poland and in the southern Baltic - expected changes and guidelines for the development of adaptation strategies in the national economy).T.1, IMGW-PIB, Warszawa, 2012 (in Polish).
  2. GUS Environment 2019. Central Statistical Office, 2019, Warszawa, www.stat.gov.pl

Line 71: the sentence “over several years now, winters in Poland have been frequently warmer than a long-term average.” needs a reference. There are some other sentences in the introduction that needs references. Please add relevant references if needed.

It has been added

Line 74-90: There is no reference here!

It has been added.

Line 140-141: What is the merit of this novelty?

In previous studies investigated drought events, the evapotranspiration in SPEI index was based on Thornthwaite PET estimation. In this study was used Penman-Monteith method.

Line 144: What do you mean in the sentence “but we also learned the occurrence of trend”?

The sentence was changed to “checked the occurrence of trend”.

Line 171: Why didn’t you obtain SPEI from the available SPEI products?

One of them: https://spei.csic.es/database.html

The SPEI base currently covers the period between January 1901 and December 2018. The aim of this study was to show how extreme was the spring drought of 2020. The SPEI Global Drought Monitor embracing the latest data, but the resolution is 1 degree and the evapotranspiration is based on Thornthwaite PET estimation. The Penman-Monteith method, which was used in this study, is considered a superior method and recommended for most uses including long-term climatological analysis.

For comparison we obtained data for grids with our stations. In the text are data for grid with Poznań, for the other stations- in Supplementary Materials.

Line 203: It would be better to check out the recent version of GLDAS which is GLDAS 2.2. If it’s possible just compare a sample of data

GLDAS 2.2 was released on 31.07.2020, when our calculations were advanced. Additionally these data started from February 2003. We used data with monthly resolutions. GLDAS 2.2  with finer spatial resolution (0.25 x 0.25) has only daily resolution (GLDAS Catchment Land Surface Model L4 daily 0.25 x 0.25 degree GRACE-DA1 V2.2 (GLDAS_CLSM025_DA1_D 2.2).

Line 243: Please increase the quality of your figures. You also need to introduce each plot separately in the caption.

It has been changed.

Line 307: Please briefly compare the trends of actual ET, potential ET, and precipitation with your SPEI result at least for a few of your stations. Use the application of the paper below:

https://www.mdpi.com/2072-4292/12/7/1221

Javadian, M.; Behrangi, A.; Smith, W.K.; Fisher, J.B. Global Trends in Evapotranspiration Dominated by Increases across Large Cropland Regions. Remote Sens. 2020, 12, 1221.

We added section about trends in ET0 and precipitation for three station: Poznań, Kalisz and Warszawa and text in discussion:

Javadian et al. (2020) demonstrated that actual evapotranspiration (AET) has significantly increased across global croplands (+14% ± 5%). Global analysis shows that the increasing trend in AET is also visible in Poland.

Line 327: Why there is a huge drop in soil moisture in the Southeastern side? Please discuss it.

The text was added.

Line 336: Please reproduce the plots. They don’t have publication quality.

It was changed.

Line 339: Why there was a moisture recovery for last winters?

Usually, there is moisture recovery in winter in Poland, as winter precipitation has been increasing with the warming climate. Precipitation is partitioned into evapotranspiration (relatively small in winter), runoff, and change of surface and soil storage that is relatively large, because harsh winters are less and less frequent nowadays, so that the ground is not frozen over much of the winter in much of Poland. Unfortunately, it is rain rather than snow. When precipitation is abounded, like in February 2020 (only rain), much of them runoff instead of percolation. Snow cover maintains moisture longer and allows for better percolation in spring.

Line 356: I would like to read the application of your research in the big picture, not just for your case study. Please add a paragraph for connecting your research to global drought studies.

The text was added:

Droughts are major weather-driven natural disasters that can occur everywhere, including water-rich areas, due to occasional anomalies in climatic variables. They can last long and encompass large areas. Drought losses have significantly increased in recent years, worldwide, for a range of reasons, including climatic and non-climatic factors [45]. In Europe, the humid north is likely to get even more humid; while the dry south is likely to get even drier [46]. As Poland is located in-between, the drought signal is more complex.

  1. Su, B.; Huang, J.; Fischer, T.; Wang, Y.; Kundzewicz, Z.W.; Zhai, J.; Sun, H.; Wang, A.; Zeng, X.; Wang, G.; Tao, H.; Gemmer, M.; Li, X.; Jiang, T. Drought losses in China might double between the 1.5°C and 2.0°C warming. Proceedings of the National Academy of Sciences (USA) PNAS, 2018, 115, 42, 10600-10605.
  2. Milly, P.C.D.; Betancourt, J.; Falkenmark, M.; Hirsch, R. M.; Kundzewicz, Z.W.; Lettenmaier, D.P.; Stouffer, R.J. Stationarity is dead: whither water management? Science, 2008, 319, 573-574.

Round 2

Reviewer 4 Report

Thank you for the revised version.

Author Response

Again, thank you for the review. We are happy that we have met your expectations.